# A Repeated Cross-Sectional Pilot Study of Physical Activity, Levels of Depression and Anxiety during the COVID-19 Pandemic in Young Greek Adults

**DOI:** 10.3390/healthcare11182493

**Published:** 2023-09-08

**Authors:** Smaragda Skalidou, Andreas Anestis, Emmanouil Skalidis, Ourania Kontaxi, Athanasia Kyrezi, Panagiota Konstantinou, Konstantinos Papadimitriou

**Affiliations:** 1Laboratory of Hygiene, Social-Preventive Medicine and Medical Statistics, Department of Medicine, School of Health Sciences, Aristotle University of Thessaloniki, 54124 Thessaloniki, Greece; sskalidou@gmail.com (S.S.); aanestis@act.edu (A.A.); manoskalidis001@gmail.com (E.S.); akyrezi@gmail.com (A.K.); 2Surgical Department, York & Scarborough Teaching Hospitals NHS Foundation Trust, York YO31 8HE, UK; ouriana97@gmail.com; 3Division of Science and Technology, The American College of Thessaloniki, 55535 Thessaloniki, Greece; 4Faculty of Life Sciences and Medicine, School of Bioscience Education, King’s College, London WC2R 2LS, UK; k21087571@kcl.ac.uk; 5Faculty of Health and Rehabilitation Sciences, University of East London, Metropolitan College of Thessaloniki, 54624 Thessaloniki, Greece

**Keywords:** COVID-19 pandemic, physical activity, mental health, depression, anxiety

## Abstract

Regular physical activity (PA) and, more specifically, exercise, is associated with lower levels of stress, depression, and anxiety. The aim of this repeated cross-sectional pilot study was to investigate the impact of participating in PA on the mental health of young adults in Greece during the COVID-19 pandemic. The study was carried out during two quarantine periods: Survey I on 5 May 2020, and Survey II on 30 April 2021. The Hamilton Anxiety (HAM-A) and Beck Depression Inventory-II (BDI-II) scales and the level of PA were used to assess a sample of individuals aged between 18 and 26 years old. In 2020 and 2021, a total of 268 (33.9% males) and 380 (37.1% females) subjects participated in the studies, respectively. According to the findings, the vast majority of the participants in both samples reported that they are physically active (*p* = 0.86), while they consider exercise as a significant health factor (*p* = 0.10). Moreover, anxiety levels statistically significant increased (*p* = 0.001), while depression levels remained relatively stable with a slight increase of approximately (*p* > 0.05). Additionally, in both surveys, individuals who engaged in a PA program exhibited reduced levels of depression and anxiety (*p* = 0.001). Also, gender appears to influence anxiety and depression levels, while a lack of exercise exacerbates these measures in both genders when compared to physically active individuals. Concludingly, it is crucial for public health strategies to include interventions that promote safe PA in the event of future lockdowns or similar emergencies.

## 1. Introduction

The novel SARS-CoV-2 coronavirus (COVID-19), which originated in Wuhan, China, in February 2020, first appeared in Greece in early March of the same year. The World Health Organization (WHO) declared COVID-19 as a global pandemic on 12 March of that year, emphasizing the importance of proactive measures to reduce the spread of the virus [1].

As a result, Greek citizens experienced two quarantine periods: the first from March to May 2020 and the second from October 2020 to May 2021, as mandated by Greek authorities. During the first quarantine period (22 March to 4 May 2020), Greece implemented a series of public health measures, including restrictions on travel and the closure of non-essential businesses.

Consequently, freedom of movement was strongly restricted, and all Greek citizens were required to stay at home to prevent the transmission of the virus. Public schools were closed, retail trade was forbidden, and social events such as marriages and christenings were postponed until the pandemic situation improved. Self-isolation was also implemented for individuals who tested positive for COVID-19 or had been in contact with COVID-19 patients [2].

The second quarantine period, lasting almost five months from 22 October 2020, continued with school closures, retail trade restrictions, and postponed social events. The restriction of movement, particularly affecting younger individuals who typically engage in outdoor activities, led to a sedentary lifestyle for several months. Consequently, daily screen time increased, while PA, such as exercise [3], significantly decreased [4]. This reduction in PA is expected to have numerous consequences on both physical and mental health, including anxiety and depression [5,6].

Anastasio et al. [7] conducted a study involving 19 patients four months after acute COVID-19 pneumonia infection. The findings revealed lower resting S_pO2_ levels and reduced S_pO2_ levels during the 6 min walk functional test. Additionally, total lung capacity, airway occlusion pressure after 0.1 s, and maximal inspiratory pressure were also diminished. Conversely, higher values were observed in the Borg scale and the modified Medical Research Council breathlessness scale compared to patients without pneumonia. These results indicate that COVID–19 is associated with lung damage, leading to a reduction in pulmonary function. Additionally, as was referenced, the impact of COVID-19 on mental health has been catastrophic. According to the WHO [8], many individuals have reported experiencing psychological distress and symptoms of depression, anxiety, or post-traumatic stress. Disturbingly, there have been indications of more widespread suicidal thoughts and behaviors, even among healthcare workers. Additionally, the closure of schools and universities has left young people vulnerable to social isolation and disconnection, amplifying feelings of anxiety, uncertainty, and loneliness, which in turn contribute to affective and behavioral issues. Furthermore, children and adolescents were confined to their homes, heightening the risk of family-related stress or abuse—both of which are significant risk factors for mental health problems [8].

In Greece, the well-being of Greek citizens has been reported as the lowest among all 27 European Union member states. As a consequence of COVID-19, the quality of life for Greek citizens worsened due to the limited PA and increased instances of eating disorders and depression symptoms during the COVID-19 lockdown [9]. It is well-established that regular PA is linked to lower levels of stress, depression, and anxiety [1]. Recent attention was directed towards PA as a potential alternative or supplement to pharmaceutical treatments for mental [1] and autoimmune disorders [10,11]. Engaging in outdoor physical activity, particularly in natural settings, has been found to positively impact mental health by promoting happiness and self-esteem while reducing stress levels [12,13].

Moreover, several studies have recognized physical activity (PA) as a coping mechanism during the COVID-19 pandemic, aiding individuals in handling stressors such as social isolation, financial challenges, and health worries. Taking part in PA was demonstrated to have positive effects in alleviating stress and enhancing mood amid this period of uncertainty [14]. While numerous studies have documented the decline in mental health among young individuals linked with lockdown periods, there is an absence of research in Greece concerning this specific age group [15]. As a result, the objective of this repeated cross-sectional pilot study is to investigate the influence of engaging in physical activity (PA) during the COVID-19 pandemic on the mental well-being of young adults (aged 18–26) in Greece. The study aims to assess alterations in PA levels and overall well-being, as indicated by anxiety and depression levels, since the commencement of the pandemic. Additionally, the study seeks to explore how gender, PA engagement status, and the perceived importance of PA for health contribute to these outcomes.

## 2. Materials and Methods

### 2.1. Study Design

The current repeated cross-sectional design was conducted with a remit to promote health and prevent illness by using anonymous online surveys at two time points. Specifically, the first survey (Survey I), which commenced on 5 May 2020, and concluded 15 of May 2020, right after the first 40 days of lockdown period had a total sample size of 363 individuals; while the second survey (Survey II), which commenced on 30 April 2021 and concluded 10 May 2021, was conducted during a long period of extraordinary restriction (six months) with a total sample size of 688 individuals. Both Surveys (I, II) were conducted close to the end of COVID–19 quarantines in Greece, aiming to maintain consistent conditions for the measurements. The participants answered the surveys within a 10-day period, while any answer after this period was excluded from the final sample.

### 2.2. Questionnaires

The 10 min survey questionnaire which was prepared using Google Forms, included the validated Greek versions of the Hamilton Anxiety (HAM-A) [16] and Beck Depression Inventory (BDI) scales [13,17,18]. The participants’ competence in PA was assessed through binary responses, “Yes” or “No” [19], while according to the WHO, their exercise frequency (at least 2 times per week), duration (75 to 150 min), intensity (low to moderate), and previous engagement in PA, before COVID-19, were also inquired about [20].

The HAM-A questionnaire comprises 14 items, which are divided into groups of symptoms related to psychological (such as mood, tension, fears, insomnia, cognitive difficulties, and low mood) and somatic (including muscular, sensory, cardiovascular, respiratory, gastrointestinal, and genitourinary symptoms) anxiety. Each symptom is rated on a scale from 0 (not present) to 4 (severe), and the total score ranges from 0 to 56. A score ≤ 17, between 18 to 24 and 25 to 30 indicates mild, mild-to-moderate, and moderate-to-severe symptoms of anxiety, respectively [16].

The BDI-II [17] is a self-report questionnaire consisting of 21 sets of statements designed to assess the levels of depression (e.g., guilt, low self-worth, irritability, and suicidal ideation). Each set is ranked in terms of severity and scored from 0 to 3. Scores of 0–9, 10–18 and 19–29 indicate a normal, a mild-to-moderate, and a moderate-to-severe range of depression. The questionnaire offers a reliable and valid index of depressive symptoms and attitudes, which can be used effectively to document changes brought about in therapy.

### 2.3. Participants

The participants were recruited through social media, websites or the authors’ email contacts. Inclusion criteria were (i) sign the informed consent, giving the permission to use the answered questions, (ii) answer all the questionnaires, (iii) aged between 18–26 years old, (iv) confirmed about their physical and mental health prior to COVID-19, (v) physical active the last one-year prior COVID-19, (vi) physical activity at least two times per week in low to moderate intensity (Borg’s scale 3 to 6 out of 10) [19] and (vii) controlled diet [20], while exclusion criteria were not the inclusion.

### 2.4. Ethics

Ethical approval for Surveys I and II was granted by the School of Medicine at Aristotle University of Thessaloniki, Greece in 2021. The study was conducted in adherence to the ethical guidelines set forth in the Helsinki Declaration of 1975, with its revised version in 2013 [20,21,22]. Furthermore, participants in the study provided online written informed consent, as the surveys were administered online. This process was designed to assure participants of the study’s anonymity and to confirm that their responses would be used solely for the purposes of the study. Additionally, the authors provided signed written consent, ensuring the participants’ willingness to take part in the study [13].

### 2.5. Statistical Analysis

The variables’ values were shown as median with standard deviation (±). Descriptive statistic and test of normality (Kolmogorov–Smirnov) (*p* ≤ 0.05) for all the variables were used for a sample of more than 50 participants [23]. Cronbach’s alpha acceptable range (0.80 ≥ α ≥ 0.76) coefficients for the used scales were estimated for both samples. The Mann–Whitney test was applied for comparisons between the cross-sectional samples and the subgroups within them. Categorical variables of “Physical activity” and the proportions of “Exercise as a significant health factor” between the two measurements were analyzed with Chi-square (χ^2^) [23]. Also, Spearman’s rank correlation coefficient (*ρ*) analysis was conducted between the positive beliefs about the significance of PA for individual health and HAM-A score. The analysis was performed with the statistical software IBM SPSS Statistics for Windows, Version 27.0. Armonk, NY, USA: IBM Corp. The level of significance was set at *a* = 0.05.

## 3. Results

The convenience sample was included by 648 subjects, who agreed to participate in the survey. In the 2020 and 2021 study, 268 (33.9% males and 63.1% females) and 380 (37.1% males and 62.9% females) subjects participated, respectively (Table 1). According to the Chi-square analysis, the vast majority of the participants in both samples (>75%) reported that they are physically active (*p* = 0.86), while even and high proportions reported that they consider exercise as a significant health factor (*p* = 0.10).

The mean score of the HAM-A Scale exhibited a significant increase (means difference: 3.3, *p* < 0.001) from 2020 to 2021, while remarkable stability was recorded regarding the BDI-II score. Additionally, in both 2020 and 2021, there was a negative correlation between positive beliefs about the significance of PA for individual health and HAM-A score (2020 vs. 2021: −0.197 vs. −0.137, *p* < 0.05). The aforementioned increase in mean HAM-A from 2020 to 2021 was also found between the same subgroups of the two samples (2020 and 2021), with its maximum degree being observed in the subgroup of low significance for PA (Table 2).

The differences between subgroups in the HAM-A scores, which were observed in the 2020 sample, remained consistent one year later. Specifically, females and non-physically active individuals had higher scores than males and physically active individuals, respectively (Table 3).

Similarly, the subgroup differences in the mean BDI-II scores observed in 2020 were also present in the 2021 sample (Table 4). Females, non-physically active individuals, and those with low perceptions of the effect of physical activity on individual health had higher scores than their counterparts.

## 4. Discussion

During the COVID-19 pandemic, the Greek government enforced strict measures that were considered some of the most stringent in Europe. These measures resulted in significant disruptions to daily life, affecting various aspects of existence. The aim of this pilot study was to assess how engaging in PA during this period (2020–2021) affected levels of depression and anxiety in Greek young adults aged 18–26 years. Furthermore, we examined possible changes in these parameters in accordance with gender and PA status. The findings of our repeated cross-sectional pilot study showed that over the course of one year, during which two lockdowns were implemented, anxiety levels (HAM-A) worsened among Greek young adults, while depression levels (BDI-II) remained stable. Gender appears to influence anxiety and depression levels, while a lack of exercise exacerbates these measures in both genders when compared to physically active individuals. Kaparounaki et al. [24] discovered that lockdowns and quarantines had a significant impact on the mental well-being of university students. The researchers noted a noteworthy rise in anxiety, depression, and suicidal thoughts among their sample of Greek university students in early April 2020, in comparison to expected scores based on the general population. Specifically, they observed a 42.5% increase in anxiety, a 74.3% increase in depression, and a 63.3% increase in overall suicidal thoughts. In line with our own findings, spanning the subsequent year (May 2020–May 2021) and due to the extension of lockdown duration, anxiety levels among this specific population appeared to be even more elevated. However, the levels of depression in our samples remained consistent across the measurements with low scores (≤13), indicating a mild depressive state [25]. This might be attributed to their ongoing engagement in physical activity (PA) with a minimum frequency of two times per week and a low-to-moderate intensity. It is widely accepted that regular PA stabilizes or reduces depression and anxiety levels, enhancing the activity of the PGC-1α/FNDC5/Irisin pathway, which in turn promotes neuronal survival [26]. However, as described, there is a broad series of studies reporting that the pandemic and subsequent lockdown measures have led to elevated levels of stress, anxiety, and mental distress among adult populations of various sociodemographic and employment-related characteristics [27] across the globe. Similarly, Varma et al. [28] found that stress, anxiety, and depression symptoms increased in 70, 59, and 39% of the participants, respectively, during the COVID-19 pandemic. However, there is no reported information about the participants’ engagement in PA. Additionally, participants with a pre-existing mental health diagnosis experienced more pronounced psychological distress. In our case, our sample reported good physical and mental health before the COVID-19 period.

Another factor that influences symptoms of stress, anxiety, and depression is the participants’ age. In our study, the participants were aged between 18 and 26 years old, and they exhibited elevated levels of anxiety during the COVID-19 period. According to the literature, age-based differences indicate that younger age groups were more susceptible to experiencing symptoms of stress, depression, and anxiety [28,29]. Additionally, among these groups, individuals in the younger adult age bracket were observed to have higher levels of perceived stress and anger compared to other age groups [30].

A study was conducted on medical staff involved in the treatment of COVID-19 patients, especially those in front-line positions, who exhibited high levels of somatization, depression, and anxiety compared to their usual reported values. They also experienced poor sleep quality and feelings of fear and terror-related to performing tasks that exposed them to uncertain conditions [31]. During lockdown periods, social interactions were negatively affected [32], and subsequent social isolation was often linked with decreased physical activity, an increased likelihood of adopting unhealthy dietary habits, and elevated levels of depression, anxiety, and stress [33].

Our pilot study identified three factors that influenced anxiety and depression scores within each cross-sectional sample: gender, PA status, and perceptions of the significance of PA for health. Apart from the higher anxiety and depression scores in females, which have been discussed elsewhere and are recently attributed to the differential expression of particular genes in the two genders [34,35], it was found that both the involvement of young individuals in physical activities and their positive perception of the role of PA in health are strongly associated with lower levels of anxiety and depression. Engaging in PA has been found to be beneficial for individuals’ overall well-being [36], whether it takes place indoors or outdoors [9].

For instance, a recent study conducted by Brailovskaia et al. [37] found that adults who maintained a consistent PA routine throughout the pandemic had lower rates of depression, anxiety, and stress, as well as higher levels of resilience, compared to their less active counterparts. Given the circumstances of the COVID-19 pandemic, PA has been proposed as a coping mechanism to potentially alleviate the detrimental impacts of the pandemic on mental health [13,38]. Interestingly, the frequency of PA per week seems to have a stronger association with well-being than its duration [9]. In our pilot study, we observed no significant change in the proportion of physically active young people between cross-sectional samples. Therefore, it is possible to hypothesize that the reduced frequency of physical activity, resulting from the implemented restrictions, led to significantly higher anxiety levels among the study population.

While our findings indicate that PA levels remained relatively stable in young adults over the course of a year of restriction measures, studies focusing on older individuals indicate a noticeable decline in PA levels. Elliot et al. [39] reported that during the third national lockdown in the UK in January 2021, PA levels among older adults significantly decreased. This decrease in PA was observed across all health conditions, age groups, neighborhood deprivation levels, and pre-pandemic activity levels. Interestingly, those who were the least active before the lockdown increased their PA during the lockdown, while those who were the most active decreased their activity levels.

In the meta-analysis by Stockwell et al. [40], out of the 66 studies that met the inclusion criteria, 64 reported changes in PA during the respective lockdowns in various populations, including children and patients with different medical conditions. The majority of studies indicated a decrease in PA and an increase in sedentary behavior during the lockdowns. Despite the numerous benefits of PA, the pandemic has created significant obstacles in maintaining a regular exercise routine. Factors such as quarantine restrictions, gym closures, and social distancing measures have limited opportunities for PA, ultimately resulting in a decline in overall activity levels among adults [41].

Our repeated cross-sectional pilot study represents the first attempt to quantitatively evaluate the combined prevalence rates of depression, anxiety, and PA among young adults in Greece during the COVID-19 pandemic. Additionally, our investigation of anxiety and depression scores associated with factors such as gender, personal perspective toward PA, and exercise frequency provides important insights into potential areas of vulnerability [42].

However, being a cross-sectional study conducted within a specific population at only two distinct points in time, it restricts the applicability of the findings to other populations or timeframes. Furthermore, it might not offer a representative portrayal of the entire population due to the participant selection process relying on online communication channels’ accessibility. Lastly, for a future relevant experimental design, the inclusion of additional demographic variables (such as educational level, socioeconomic status, or health status), as well as considerations of factors like sleep quality [43], dietary patterns [44], and feelings of loneliness [45], would contribute to a more comprehensive comprehension of the elements that contribute to mental health status. This would encompass individuals’ engagement in physical activity and their personal perspectives on the matter. The findings of this study could hold considerable significance in identifying the support needs of young individuals or other populations susceptible to mental health challenges during potential future severe health crises. They could also play a pivotal role in designing targeted interventions to enhance resilience and decrease vulnerability in pandemic scenarios. The noteworthy impact of the COVID-19 pandemic on the mental well-being of young individuals has prompted global concern among policymakers and practitioners. Consequently, it is imperative to explore innovative strategies that can enhance access to mental health services and foster mental wellness.

For instance, maintaining social connections despite isolation and revitalizing social bonds during the recovery phase could prove to be effective measures. Additionally, recognizing and offering assistance to young adults disproportionately affected, coupled with implementing broader preventive measures, may help alleviate the potential long-term mental health repercussions for this susceptible demographic.

## 5. Conclusions

The prolonged lockdown periods in Greece during the COVID-19 pandemic have resulted in a decline in anxiety levels among young adults, especially those with low levels of PA who perceive PA as having minimal impact on their health. Considering the numerous health advantages associated with increased PA and decreased sedentary behavior, it becomes imperative for public health strategies to include interventions that promote safe physical activity in the case of future lockdowns or similar emergencies. Governments and mental health agencies should establish effective monitoring and screening programs to swiftly and systematically identify vulnerable subpopulations.

## 6. Study Limitations

Based on the study’s timeframe and exclusion criteria, several limitations are evident:

A. While we employed a G*Power sample size calculation to determine the required sample size, the implementation of multiple exclusion criteria resulted in a reduction of our final participant count. Consequently, a pilot study was undertaken, and further investigation is warranted.

B. There exists the possibility of biases in self-reporting based on participants’ engagement in physical activity. Utilizing the International Physical Activity Questionnaire (IPAQ) or Global Physical Activity Questionnaire (GPAQ) can provide a more validated depiction of participants’ physical activity levels.

C. In our study, we only utilized the variables of gender, age, and participation in physical activity (PA). For future studies, it is imperative to incorporate a broader range of variables, including socioeconomic status, a wider age range of participants (potentially comparing young and older individuals), diverse forms of exercise (such as dry-land, aquatic, aerobic, or anaerobic exercises), and living conditions (such as feelings of loneliness, single parenting, or household composition with multiple children, etc.).

## Figures and Tables

**Table 1 healthcare-11-02493-t001:** Demographic characteristics of the study populations.

Physical Activity Exercise	First Measurement 2020		Second Measurement 2021	
	n	%	n	%
Yes	205	76.5%	293	77.1%
No	63	23.5%	87	22.9%
Do you consider physical activity significant for a person’s health? (0 = not at all, 4 = absolutely)				
	n	%	n	%
Not at all	3	1.1%	2	0.5%
Probably not	10	3.7%	16	4.2%
Neither yes, nor no	55	20.5%	54	14.2%
Probably yes	89	33.2%	115	30.3%
Absolutely yes	111	41.4%	193	50.8%

**Table 2 healthcare-11-02493-t002:** Comparison of the psychometric test scores in the cross-sectional samples (total population).

	First Measurement 2020	Second Measurement 2021	
	Mean (SD)	Mean (SD)	*p*-Value
HAM-A score	14.5 (7.0)	17.8 (10.0)	0.001 *
BDI-II score	11.7 (7.7)	11.9 (9.7)	0.232

*: Significant difference. Comparison was conducted with Mann–Whitney U test.

**Table 3 healthcare-11-02493-t003:** Comparison of the HAM-A psychometric test scores in the subgroups of the cross-sectional samples.

	First Measurement 2020			Second Measurement 2021		
	n	Mean (SD)	*p*-Value	n	Mean (SD)	*p*-Value
Males	91	11.6 (5.8)	0.001 *	141	14.2 (9.6)	0.001 *
Females	177	16.1 (7.0)		239	20.0 (9.7)	
Physical activity Exercise						
Yes	205	13.8 (6.8)	0.001 *	293	16.7 (9.5)	0.001 *
No	63	17.1 (7.0)		87	21.7 (10.8)	
Perceived significance of physical activity for individual health						
Low/moderate low	13	18.3 (6.5)	0.015 *	18	25.4 (8.9)	0.014 *
Moderate	55	15.9 (7.1)		54	20.8 (9.4)	
Moderate high/high	200	13.9 (6.9)		308	17.1 (9.9)	

*: Significant difference. Comparison was conducted with Mann–Whitney U test.

**Table 4 healthcare-11-02493-t004:** Comparison of the BDI-II psychometric test scores in the subgroups of the cross-sectional samples.

	First Measurement 2020		Second Measurement 2021			
	n	Mean (SD)	*p*-Value	n	Mean (SD)	*p*-Value
Males	91	10.1 (6.93)	0.010 *	141	10.5 (9.39)	0.007 *
Females	177	12.5 (7.85)		239	12.7 (9.87)	
Physical activity						
Yes	205	10.8 (6.98)	0.020 *	293	11.0 (9.35)	0.002 *
No	63	14.6 (8.88)		87	14.7 (10.5)	
Perceived significance of physical activity (exercise) for individual health						
Low/moderate low	13	18.6 (5.04)	0.001 *	18	20.1 (7.12)	0.001 *
Moderate	55	12.9 (8.30)		54	15.1 (9.76)	
Moderate high/high	200	11.0 (6.64)		308	10.8 (9.23)	

*: Significant difference. Comparison was conducted with Mann–Whitney U test.

## Data Availability

The authors confirm that the data supporting the findings of this study are available within the article.

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
