# Peer review of "A Repeated Cross-Sectional Pilot Study of Physical Activity, Levels of Depression and Anxiety during the COVID-19 Pandemic in Young Greek Adults"

_healthcare, 2023, doi:10.3390/healthcare11182493_

Round 1

Reviewer 1 Report

Comments to the Author

The authors of this paper report some interesting data aiming to investigate the impact of participating in physical activity (PA) on the mental health of young adults in Greece during the COVID-19 pandemic. However, there are several points that require further clarity;

1- Page 1, Lines 34-79: I recommend that you add a paragraph somewhere in this section that talks about the physiological as well as psychological damages of COVID-19 infection. This impact has been especially notable within the athlete population, further emphasizing the far-reaching consequences of the virus. The heightened concerns surrounding the transmission of a potentially fatal or enduringly harmful infection have prompted widespread adoption of preventive measures like lockdowns and quarantines. Therefore, these measures, while essential for curtailing the spread of the virus, have inevitably led to detrimental psychological outcomes. For more in-depth insights, please consult the references provided below.

https://doi.org/10.1016/j.resp.2022.103983

DOI: 10.1183/13993003.04015-2020

2- Page 2, Line 81: The rationale behind the selection of these specific time points remains unclear. I do not see a factor between the two-time points that would influence the results. Notably, both time points correspond to distinct shutdown phases within the pandemic timeline.

Furthermore, the exact time frames from which data during the lockdown periods were extracted lack clarity; specifics such as whether the data originated from, for instance, the second or fifth week of lockdown, remain unspecified.

3- Page 2, Line 89: More details about the recruitment process of study participants should be provided to exclude a serious selection bias. This should encompass a detailed breakdown of participants' attributes, including their chronic psychological or physiological health status, prior occurrences of COVID-19 infection, any pertinent history in the realm of sports involvement (including sports background status/year, if any), as well as additional factors with the potential to exert a substantial influence on the research outcomes.

In addition, to accurately portray participants' physical activity levels or status, I recommend incorporating a validated physical activity assessment scale, such as a suitable International Physical Activity Questionnaire (IPAQ). If you get a yes or no response without informing the participants about which activities are physical activity and under what conditions and for how long, this can lead to a big misunderstanding. Please provide more details about the type, frequency and intensity of activity of the group doing physical activity (in the participants section of the method).

4- Page 3, Line 96: Please provide the crombah alpha values obtained from your data for both scales.

5- Page 3, Lines 115-116: Please indicate the number and year of the ethics committee report. Please also provide these details in the "Institutional Review Board Statement" section below.

6- Page 3, Line 122: Did the authors perform some power and sample size calculations? This is especially important as the numbers are pretty small.

7- Page 4, Lines 145-147: As far as I understand, the datasets collected in 2020 and 2021 seem to encompass distinct participant cohorts. Therefore, comparisons between years of data do not reflect absolutely and absolutely accurate results. Because there are dozens of environmental, individual, and even genetic factors that determine psychological conditions such as anxiety and depression. I believe that the hypothesis is not established correctly and the method used is wrong. Please reconsider your methods and hypotheses. Remove this table and carefully revise the relevant lines from the abstract to the conclusion (including the tables).

8- Pages 4-5, Lines 152-163: Please remove the %change row in Tables 3 and 4. This makes no sense for the reasons I mentioned above (different participants in 2020 and 2021).

GENERAL COMMENTS:

1. The topic is important but especially the introduction and discussion sections should be improved significantly. Literature review is pretty nonadequacy.

2. Abstract should be re-edited after changes made in the article.

The language is fine.

Author Response

Reviewer 1

Dear reviewers

We really appreciate your comments and quick response to the judge of our manuscript. Your indications helped us to improve the manuscript.

C: Comment

A: Answer

  1. C. 1- Page 1, Lines 34-79: I recommend that you add a paragraph somewhere in this section that talks about the physiological as well as psychological damages of COVID-19 infection. This impact has been especially notable within the athlete population, further emphasizing the far-reaching consequences of the virus. The heightened concerns surrounding the transmission of a potentially fatal or enduringly harmful infection have prompted widespread adoption of preventive measures like lockdowns and quarantines. Therefore, these measures, while essential for curtailing the spread of the virus, have inevitably led to detrimental psychological outcomes. For more in-depth insights, please consult the references provided below.

https://doi.org/10.1016/j.resp.2022.103983

DOI: 10.1183/13993003.04015-2020

  1. You are absolutely right. We added two paragraphs analyzing extensively the impact of Covid – 19 in both physical and mental health. Lines: 58 – 75.

C . 2- Page 2, Line 81: The rationale behind the selection of these specific time points remains unclear. I do not see a factor between the two-time points that would influence the results. Notably, both time points correspond to distinct shutdown phases within the pandemic timeline. Furthermore, the exact time frames from which data during the lockdown periods were extracted lack clarity; specifics such as whether the data originated from, for instance, the second or fifth week of lockdown, remain unspecified.

  1. We understand your questions and we answered them. Lines 107 – 110.

C . 3- Page 2, Line 89: More details about the recruitment process of study participants should be provided to exclude a serious selection bias. This should encompass a detailed breakdown of participants' attributes, including their chronic psychological or physiological health status, prior occurrences of COVID-19 infection, any pertinent history in the realm of sports involvement (including sports background status/year, if any), as well as additional factors with the potential to exert a substantial influence on the research outcomes.      In addition, to accurately portray participants' physical activity levels or status, I recommend incorporating a validated physical activity assessment scale, such as a suitable International Physical Activity Questionnaire (IPAQ). If you get a yes or no response without informing the participants about which activities are physical activity and under what conditions and for how long, this can lead to a big misunderstanding. Please provide more details about the type, frequency and intensity of activity of the group doing physical activity (in the participants section of the method).

  1. We included the details that you suggested. Details about physical activity (duration, intensity, and repeatability) were recorded by the participants. We included the reference that we supported our methodology about the participants’ recruitment. Also, we included the lack of IPAQ as a limitation of our study and a good idea for the next step of the study. Lines 143 – 149 & 329-331.

Papadimitriou, K.; Detopoulou, P.; Soufleris, K.; Voulgaridou, G.; Tsoumana, D.; Ntopromireskou, P.; Giaginis, C.; Chatziprodromidou, I.P.; Spanoudaki, M.; Papadopoulou, S.K. Nutritional Risk and Sarcopenia Features in Patients with Crohn’s Disease: Relation to Body Composition, Physical Performance, Nutritional Questionnaires and Biomarkers. Nutrients 202315, 3615. https://doi.org/10.3390/nu15163615.

  1. 4- Page 3, Line 96: Please provide the crombah alpha values obtained from your data for both scales.
  2. Done. Line 163.
  3. 5- Page 3, Lines 115-116: Please indicate the number and year of the ethics committee report. Please also provide these details in the "Institutional Review Board Statement" section below.
  4. Ethical approval for Surveys I and II was granted by the School of Medicine at Aristotle University of Thessaloniki, Greece in 2021 (Line 152). The study was conducted in adherence to the ethical guidelines set forth in the Helsinki Declaration of 1975, with its revised version in 2013. Also, the authors provided signed written consent, ensuring the participants' willingness to take part in the study. Please, check the reference below. Also, we included that statement below as you suggested (Lines 342-343).

Debowska, A., Horeczy, B., Boduszek, D., & Dolinski, D. (2022). A repeated cross-sectional survey assessing university students' stress, depression, anxiety, and suicidality in the early stages of the COVID-19 pandemic in Poland. Psychological Medicine, 52(15), 3744-3747. doi:10.1017/S003329172000392X

C . 6- Page 3, Line 122: Did the authors perform some power and sample size calculations? This is especially important as the numbers are pretty small.

A . We recognize the significance of G* power analysis as a vital methodological tool. Nonetheless, the application of multiple exclusion criteria led to a reduction in our final number of participants. In Survey I, the initial count of 363 participants decreased to 268, and in Survey II, the original 688 participants decreased to 380. Furthermore, no additional participants expressed an interest in participating. Therefore, we decided to conduct a pilot study. Consequently, we have incorporated a limitations section at the conclusion of the manuscript (Lines 325-328).

  1. 7- Page 4, Lines 145-147: As far as I understand, the datasets collected in 2020 and 2021 seem to encompass distinct participant cohorts. Therefore, comparisons between years of data do not reflect absolutely and absolutely accurate results. Because there are dozens of environmental, individual, and even genetic factors that determine psychological conditions such as anxiety and depression. I believe that the hypothesis is not established correctly and the method used is wrong. Please reconsider your methods and hypotheses. Remove this table and carefully revise the relevant lines from the abstract to the conclusion (including the tables).
  2. The present study encompasses distinct participant cohorts. This is the reason that we conducted a repeated cross-sectional pilot study. Repeated cross-sectional study is utilized in different cohorts that can be used to consider patterns of change at the aggregate level (e.g. we want to observe the percentage of smokers between 2000 and 2020). In addition, trying to reduce the imponderable factors such as environmental, individual, and even genetic factors, we used strict inclusion criteria ( i) sign the informed consent, giving the permission to use the answered questions, ii) answer all the questionnaires, iii) aged between 18 – 26 years old, iv) ensured about physical and mental health prior COVID – 19, v) physical active the last one-year prior COVID – 19, vi) physical activity at least two times per week in low to moderate intensity (Borg’s scale 3 to 6 out of 10) and vii) controlled diet; while the exclusion criteria were not the inclusion). They have conducted several repeated cross-sectional studies similar to ours. We provide two references below:

Debowska, A., Horeczy, B., Boduszek, D., & Dolinski, D. (2022). A repeated cross-sectional survey assessing university students' stress, depression, anxiety, and suicidality in the early stages of the COVID-19 pandemic in Poland. Psychological Medicine, 52(15), 3744-3747. doi:10.1017/S003329172000392X.

Banić, A., Sapunar, D., & Buljan, I. (2023). Changes in information-seeking patterns and perception of health crisis management in a year of COVID-19 pandemic: a repeated cross-sectional study. Croatian medical journal, 64(2), 93–102. https://doi.org/10.3325/cmj.2023.64.93.

C .Pages 4-5, Lines 152-163: Please remove the %change row in Tables 3 and 4. This makes no sense for the reasons I mentioned above (different participants in 2020 and 2021).

A . With all the respect, as we discussed above this is the nature of that kind of study. A repeated cross–sectional data refers to data where a new random sample is collected at successive surveys. There are several examples where the authors confide the percentages of the participants at each time point. Please, check the well-written guidance about cross-sectional studies.

GENERAL COMMENTS:

C . The topic is important but especially the introduction and discussion sections should be improved significantly. Literature review is pretty nonadequacy.

A . According to your helpful suggestions we provided improvements in the whole manuscript.

C . Abstract should be re-edited after changes made in the article.

A . Done

Reviewer 2 Report

General Comments: This report on the findings from two cross sectional surveys during the COVID-19 pandemic (3/2020-5/2020; 4/2021-5/2021 among Greek young adults is both interesting and potentially informative.  The authors seek to describe the associations between self-reported physical activity and perceived health benefit of physical activity among the respondents.  The exposure variable for physical activity is apparently a dichotomous variable constituting, "Yes, I'm physically Active" or "No, I'm not physically active".  The measures for anxiety and depression are standard instruments with well documented validity and reliability measures for this study population.  Further explanation for the measures of physical activity need to be provided and are currently missing from the current manuscript.  The authors cite others who have measured physical activity behaviors in previous studies where valid and reliable measures for self-report physical activity have been used and well documented within the narrative of each of these studies (References 1 and 6).  Hence, further assessment of the findings and conclusions from this study cannot be completed without knowing about what measures were used in the assessment of physical activity among all respondents.  The current estimates of reported physical activity >70% among these young adults appears rather skewed relative to national Greek estimates as reported in the Global Physical Activity Observatory.  How is your term 'physically active' for these respondents determined?  What definition of physical activity are you using among these respondents, those that meet the WHO minimal levels for physical activity of 150 minutes per week or more, or the response to a simple dichotomous question?  If the latter then issues of selection bias need to be addressed. 

Author Response

Reviewer 2

Dear reviewers

We really appreciate your comments and quick response to the judge of our manuscript. Your indications helped us to improve the manuscript.

C: Comment

A: Answer

C . Further explanation for the measures of physical activity need to be provided and are currently missing from the current manuscript. The authors cite others who have measured physical activity behaviors in previous studies where valid and reliable measures for self-report physical activity have been used and well documented within the narrative of each of these studies (References 1 and 6).  Hence, further assessment of the findings and conclusions from this study cannot be completed without knowing about what measures were used in the assessment of physical activity among all respondents. The current estimates of reported physical activity >70% among these young adults appears rather skewed relative to national Greek estimates as reported in the Global Physical Activity Observatory.  How is your term 'physically active' for these respondents determined?  What definition of physical activity are you using among these respondents, those that meet the WHO minimal levels for physical activity of 150 minutes per week or more, or the response to a simple dichotomous question?  If the latter then issues of selection bias need to be addressed. 

A . Additional information about the measurement of physical activity has been included. We have included the reference that our Physical activity detection regarded. Furthermore, we have included additional questions that clarify the level of Physical activity (Lines 143 – 149). However, we have also acknowledged it as a limitation of our study, which could potentially be addressed in future research Lines 329 – 331.

Papadimitriou, K.; Detopoulou, P.; Soufleris, K.; Voulgaridou, G.; Tsoumana, D.; Ntopromireskou, P.; Giaginis, C.; Chatziprodromidou, I.P.; Spanoudaki, M.; Papadopoulou, S.K. Nutritional Risk and Sarcopenia Features in Patients with Crohn’s Disease: Relation to Body Composition, Physical Performance, Nutritional Questionnaires and Biomarkers. Nutrients 2023, 15, 3615. https://doi.org/10.3390/nu15163615.

Round 2

Reviewer 1 Report

I appreciate your willingness to consider my suggestions. While I believe that your efforts have improved the study, there are some points that need to be revised in relation to items 7 and 8 that I presented in the previous report.

The first study you cited examines depression, anxiety, and stress levels in both male and female subjects across multiple waves, employing a mixed-design approach. However, the methodology is flawed for this type of study, which seeks to compare perception/attitude metrics across different populations. It should not be overlooked that even reputable journals may have reviewers who do not have the statistical expertise to detect such errors.

On the other hand, the second study used chi-squared to determine whether the categorical variables differed by years (2020-2021). This may be a correct method because, like the example you gave above (e.g. we want to observe the percentage of smokers between 2000 and 2020), the authors are trying to determine the percentage participation values of xxxx variables between years. I want to clarify that my critique is focused on the statistical methods used, rather than the overall approach of conducting repeated cross-sectional studies. Methods like the mixed-design are less reliable when examining factors subject to individual variability, such as anxiety and depression.

In conclusion, I strongly recommend either removing the table, as it's not compatible with the employed statistical methods, or revising it using a more appropriate test, such as chi-square.

Author Response

Reviewer 1

  1. The first study you cited examines depression, anxiety, and stress levels in both male and female subjects across multiple waves, employing a mixed-design approach. However, the methodology is flawed for this type of study, which seeks to compare perception/attitude metrics across different populations. It should not be overlooked that even reputable journals may have reviewers who do not have the statistical expertise to detect such errors.

On the other hand, the second study used chi-squared to determine whether the categorical variables differed by years (2020-2021). This may be a correct method because, like the example you gave above (e.g. we want to observe the percentage of smokers between 2000 and 2020), the authors are trying to determine the percentage participation values of xxxx variables between years. I want to clarify that my critique is focused on the statistical methods used, rather than the overall approach of conducting repeated cross-sectional studies. Methods like the mixed-design are less reliable when examining factors subject to individual variability, such as anxiety and depression.

I strongly recommend either removing the table, as it's not compatible with the employed statistical methods, or revising it using a more appropriate test, such as chi-square.

  1. Thank you very much for your consistency in improving our article. As you suggested we deleted the percentages in Tables 2, 3, and 4. Also, the Categorical variables of “Physical activity” and the proportions of “Exercise as a significant health factor” between the two measurements were analyzed with Chi-square (χ2). Finally, we modified the abstract, results, and discussion.

Reviewer 2 Report

The manuscript is much improved.  The anxiety and depression scales are appropriately used and evidence of both reliability (repeatability) and validity have been noted by appropriate citations. The role in of physical activity, and specifically exercise which is planned and intentional is further defined.  Although a rather weak dichotomous measure, which the authors acknowledge, the exposure variable here is the perception of the role of 'exercise' in coping with potentially increased anxiety and depression during the COVID-19 lockdowns, which is satisfactory for such a study.  However, should the authors wish to pursue the association of reported physical activity and its effects on anxiety and depression, it is recommended that the authors use a standardized physical activity survey instrument such as the International Physical Activity Questionnaire (IPAQ) or the Global Physical Activity Questionnaire (GPAQ), as these are valid and reliable instruments and are available in Greek and many other languages.  As a pilot study, the current manuscript provides some important insights into the perception of young adults during the pandemic and the role of physical activity as one of several coping strategies during such an austere period of time.

Author Response

Reviewer 2

  1. The manuscript is much improved.  The anxiety and depression scales are appropriately used and evidence of both reliability (repeatability) and validity have been noted by appropriate citations. The role in of physical activity, and specifically exercise which is planned and intentional is further defined.  Although a rather weak dichotomous measure, which the authors acknowledge, the exposure variable here is the perception of the role of 'exercise' in coping with potentially increased anxiety and depression during the COVID-19 lockdowns, which is satisfactory for such a study.  However, should the authors wish to pursue the association of reported physical activity and its effects on anxiety and depression, it is recommended that the authors use a standardized physical activity survey instrument such as the International Physical Activity Questionnaire (IPAQ) or the Global Physical Activity Questionnaire (GPAQ), as these are valid and reliable instruments and are available in Greek and many other languages.  As a pilot study, the current manuscript provides some important insights into the perception of young adults during the pandemic and the role of physical activity as one of several coping strategies during such an austere period of time.
  2. We understand your concern and point in this comment. Observing the methodology, we tried to ensure the minimum requirements of a physical active sample. Therefore, according to the World Health Organization (WHO), we included their exercise frequency (at least 2 times per week), duration (75 to 150min), intensity (low to moderate), and previous engagement in PA (Lines 119-123). Moreover, in the limitations section, we included the Global Physical Activity Questionnaire (GPAQ) as you suggested and it is a good idea for our second round of measurements.